# Robust Adamantane-Based Membranes with Enhanced Conductivity for Vanadium Flow Battery Application

**DOI:** 10.3390/polym14081552

**Published:** 2022-04-11

**Authors:** Bengui Zhang, Xueting Zhang, Qian Liu, Yanshi Fu, Zhirong Yang, Enlei Zhang, Kangjun Wang, Guosheng Wang, Zhigang Zhang, Shouhai Zhang

**Affiliations:** 1College of Chemical Engineering, Shenyang University of Chemical Technology, Shenyang 110142, China; mztkn5y4r@163.com (X.Z.); 13470013906@163.com (Y.F.); yzr1642000898@163.com (Z.Y.); zhangel1979@126.com (E.Z.); wgsh-lyc@163.com (G.W.); zhzhgang@126.com (Z.Z.); 2State Key Laboratory of Fine Chemicals, School of Chemical Engineering, Dalian University of Technology, Dalian 116024, China; cavenlouis@outlook.com

**Keywords:** vanadium redox flow battery, adamantane-based, ion conductivity, area resistance, selective swelling

## Abstract

Membranes with high conductivity, high selectivity, and high stability are urgently needed for high-power-density vanadium flow batteries (VFBs). Enhancing membrane conductivity presents many challenges, often resulting in sacrificing membrane selectivity and mechanical strength. To overcome this, new robust adamantane-based membranes with enhanced conductivity are constructed for VFB. Low-content basic piperazine (IEC = 0.78 mmol g^−1^) and hydrophilic hydroxyl groups are introduced into highly rigid, hydrophobic adamantane containing poly(aryl ether ketone) backbone (PAPEK) and then selectively swelled to induce microphase separation and form ion transport pathways. The highly rigid and hydrophobic PAPEK exhibits high swelling resistance and provides the membranes with slight swelling, high selectivity, and high mechanical strength. The selective swelling temperature has a significant influence on the areal resistance of the resulting membrane, e.g., the PAPEK-130 membrane, when selectively swelled at 130 °C, has low areal resistance (0.22 Ω∙cm^2^), which is approximately two-fifths that of the PAEKK-60 membrane (treated at 60 °C, 0.57 Ω∙cm^2^). Consequently, the resulting PAPEK membranes exhibit low swelling, high selectivity, and low areal resistance, with the VFB constructed with a PAPEK-90 membrane exhibiting excellent energy efficiency (91.7%, at 80 mA∙cm^−2^, and 80.0% at 240 mA∙cm^−2^) and stable cycling performance for 2000 cycles.

## 1. Introduction

Large-scale energy storage technology has essential application prospects in peak shaving and valley filling of power grids and solving fluctuations in renewable energy sources, such as wind and solar energy [1,2]. The intermittent nature of renewable energy sources will open spatial and temporal gaps between the availability of the energy and its consumption by end users [3]. Many flow battery technologies are suitable for large-scale energy storage, such as zinc-based [4], iron-based [5], and vanadium flow batteries (VFB). VFB technology enjoys high safety and long life and is one of the most promising technologies in large-scale electrochemical energy storage [6,7]. In a VFB, the membrane plays a crucial role in separating positive and negative electrolytes, preventing the cross-mixing of vanadium (V) ions with different valence states, and conducting ions to complete the internal circuit [8]. Perfluorosulfonic acid membranes, such as Nafion membranes, have been extensively studied for VFB due to their high ion conductivity and chemical stability. However, the cost of Nafion membranes is very high, and the membrane selectivity is poor, which severely hinders their wide application in VFBs. Advanced membranes, with high performance and low cost, have been developed based on various membrane materials [9,10,11,12,13,14,15,16,17,18,19,20,21,22,23,24,25,26,27], and many membrane preparation methods have been developed, such as porous membranes [22], and inorganic-organic membranes [28]. These studies have contributed to the rapid development of advanced membranes for VFBs.

Among these advanced membranes, positively charged membranes enjoy high membrane selectivity benefits from the Donnan repulsion effect [29], and have received widespread attention [13,15,16,17,18,19,20,23,26,27,30,31,32,33]. However, the membrane conductivity of positively charged membranes needs to be further improved to meet the requirements of high-power-density VFB, which has significant advantages in reducing the volume and cost of battery systems [7,8]. Various advanced membranes [4,30,31,32] have been developed to overcome these shortcomings, which reportedly have significantly improved membrane conductivity. However, many of these membranes still present challenges, including the high complexity of controlling the microstructure in the preparation process of porous membranes [19,21,34], wide pore size distribution [35], and relatively low mechanical strength due to porous membrane structure.

Generally, the microphase separation structure formed in a membrane significantly increases membrane conductivity [21,36,37]. The typical strategy has been to improve microphase separation by increasing membrane IEC or introducing comb-shaped chains or block polymers [38]. However, some significant challenges remain, such as increasing membrane IEC, resulting in severe membrane swelling, decreased mechanical strength, and impaired membrane selectivity [29]. Additionally, membranes based on comb-shaped or block polymers often have complex synthetic processes [38]. The swelling-induced phase separation is an available strategy, which has been used to prepare membranes with significantly enhanced conductivity [32,39,40,41,42]. However, challenges such as the balance between membrane swelling and selectivity need to be further improved [39,42].

In this study, for the first time, low-content piperazine and hydroxyl groups were introduced into highly swelling-resistant adamantane-containing poly(aryl ether ketone) (PAPEK) backbone and then selectively swelled to form ion transport pathways in the membrane (Figure 1). In this design, a basic piperazine group can offer two basic sites with acid–basic interaction capability, increasing the membrane’s acid doping level (ADL) and benefiting the membrane conductivity. In addition, hydrophilic hydroxyl groups can form H-bonds with acids due to the affinity between hydroxyls and acid, which would benefit membrane conductivity. Furthermore, the protonated piperazine groups (positively charged) have the Donnan effect with vanadium ions, enabling the membrane to have high selectivity. Then, the membrane segments containing piperazine and hydroxyl groups were selectively swelled in hot H_3_PO_4_ (85 wt.%) to form highly conductive ion transport pathways. Moreover, the highly rigid adamantane-based PAPEK backbone possessed high swelling resistance [16,43,44], which would provide the membrane with robust mechanical strength and high selectivity. The resulting PAPEK membranes exhibited excellent VFB performance, fairly close to the highest values ever reported [7]. Herein, the selective swelling behaviors, membrane properties, and the battery performance of PAPEK membranes were studied.

## 2. Materials and Methods

The information regarding materials used is summarized in the Appendix A.

### 2.1. Preparation of the PAPEK Membranes

Chloromethylated adamantane, containing poly(aryl ether ketone) (CAPEK) with a low degree of chloromethylation (DC) at 0.5 (1.0 g), was dissolved in 1-methyl-2-pyrrolidinone (NMP, 7.3 mL) and *N*-2-hydroxyethyl piperazine (0.5 g) and then added to the solution and stirred at room temperature for 24 h. Next, the solution was centrifuged and spread on a clean horizontal glass pane with a doctor blade (500 μm) and dried at 70 °C for 4 h. Finally, the glass pane was dipped in deionized water to peel off the obtained membrane (PAPEK-virgin), with membrane thickness at 40–45 μm. These membranes were selectively swelled with phosphoric acid (H_3_PO_4_, 85 wt.%) at 60, 90, or 130 °C for 4 h, after which the membranes were washed, exchanged with sulfuric acid (H_2_SO_4_, 3 M, 24 h), and then stored in new H_2_SO_4_ solution (3 M). The resulting membranes (PAPEK-60, 90, and 130) were named according to the swelling temperature.

### 2.2. Characterization Methods

Methods used to characterize PAPEK membranes, including Fourier transform–infrared spectroscopic (FT-IR) analysis, small-angle X-ray scattering (SAXS), acid doping level, mechanical strength, swelling ratio, area resistance, permeability, battery performance, and ex situ stability tests, are summarized in the Appendix A.

## 3. Results and Discussion

The APEK and CAPEK exhibited good solubility in some solvents, such as chloroform, *N*-methyl pyrrolidone, dichloromethane, and *N*-dimethyl acetamide (Table 1). ^1^H NMR analysis of adamantane-based poly(aryl ether ketone) (APEK) and CAPEK was carried out using chloroform-d as the solvent (Figure 1a, Appendix A), and the degree of substitution (DS) was determined using ^1^H NMR [16,45,46]. The CAPEK with DS = 0.5 was used to prepare the PAPEK membrane in this work (Figure 1). The molecular weight (M_n_ = 4.2 × 10^4^) of the APEK [41] was determined using gel permeation chromatography (GPC) (Appendix A). The FT-IR spectra of CAPEK and PAPEK membranes are summarized in Figure 1b. The characteristic absorption peaks, attributed to poly(aryl ether ketone)s [42,47], at 1648 (C=O) and 1244 cm^−1^ (C-O-C) were observed in the CAPEK and PAPEK membrane spectra. For the PAPEK-virgin membrane, new absorption peaks at 3485 and 2810 cm^−1^ were observed, probably attributable to absorptions of −OH and C-H of *N*-2-hydroxyethyl piperazine, respectively, which suggests that *N*-2-hydroxyethyl piperazine was attached in the membrane. A strong, wide absorbance band at ~3399 cm^−1^ was viewed for the PAPEK-130 membrane due to −OH absorbance of H_2_SO_4_/H_2_O [40,48] and −OH of *N*-2-hydroxyethyl piperazine groups.

The ion exchange capacity (IEC) of the PAPEK membrane was determined by titration, and the IEC for PAPEK-membrane (Cl form) was 0.78 mmol g^−1^. One unit of protonated piperazine group would bind to two units of Cl ions. Therefore, the piperazine content of the PAPEK membrane was half the IEC of the PAPEK membrane (Cl form). Therefore, in this work, the hydroxyethylpiperazine content in the PAPEK membrane was 0.39 mmol g^−1^.

The TGA of APEK, CAPEK and DSC of APEK [16] are shown in Appendix A; the T_g_ for APEK and CAPEK were 260 °C and 237 °C, respectively. The high T_g_ of adamantane-containing backbone was mainly due to the rigid, bulky adamantane constrained motion of the polymer chain [44]. The high rigidity adamantane-containing backbone would limit membrane swelling and maintain the high mechanical strength of the PAPEK membrane. TGA thermograms of APEK, CAPEK were displayed in Appendix A. APEK and CAPEK exhibited high thermal stability [16], and the maximum temperature of the swelling treatment used in this work was 130 °C. Therefore, the adamantane-containing backbone had sufficient thermal stability to cope with this work.

The PAPEK-virgin membrane exhibited good thermal stability (Figure 2), as evidenced by the high degradation temperature of about 210 °C. The weight loss of the PAPEK-virgin membrane between 220 °C and 360 °C was probably due to the loss of morpholine groups in the membrane. Compared to the PAPEK-virgin membrane, the PAPEK-90 membrane exhibited weight loss (~1.6%) in the range before 110 °C, probably due to water loss from the membrane. This result showed that water was absorbed in the membrane after the PAPEK membrane was swollen with phosphoric acid, indirectly indicating that phosphoric acid entered the membrane. The TGA data also showed that the PAPEK membrane was stable below 190 °C, and thus its thermal stability met the VRFB requirements.

Generally, microphase-separated structures formed in membranes can significantly enhance membrane conductivity [36]. In this study, nanophase-separated structures were induced by the selective swelling process using high-concentration H_3_PO_4_ (85 wt.%). Because the basic piperazine and hydrophilic hydroxyl grafted in PAPEK membranes had acid–base interactions and formed H-bonds with H_3_PO_4_, respectively, these membranes could be selectively swelled by H_3_PO_4_. Meanwhile, the highly rigid, bulky hydrophobic PAPEK backbone offered high swelling resistance, thus imparting the membrane robust mechanical stability and high selectivity. The acid doping level (ADL) was expressed as the number of moles of acid per polymer repeat unit [49]. The ADL of PAPEK membranes in H_3_PO_4_ was affected by the selective swelling temperature, which varied from 1.2 to 2.0 (Figure 3a). The ADL of PAPEK membranes in sulfuric acid increased with the increase of the selective swelling treatment temperature in phosphoric acid, because the selective swelling process in phosphoric acid caused the PAPEK membrane to swell, and the swelling increased with increasing temperature (Figure 3b). Therefore, after the phosphoric acid in the membrane was exchanged with sulfuric acid, the membrane also exhibited an increased ADL in sulfuric acid.

The PAPEK-130 membrane swelled at 130 °C exhibited the highest ADL of 2.0, a significantly lower ADL than that of many reported PBI membranes in high-concentration H_3_PO_4_ [50]. These results were mainly because, on the one hand, PAPEK membranes were made from CAPEK with low functionalization of DC at 0.5. Consequently, the derived PAPEK membranes had low piperazine content. On the other hand, the highly rigid and hydrophobic adamantane-containing main chain resulted in high steric hindrance, which restricted excessive membrane swelling [15,16,43] and limited absorbed free acids due to H-bonding effects [51].

The swelling ratio of PAPEK membranes showed that these membranes exhibited high swelling resistance in H_3_PO_4_ (85 wt.%), with a swelling ratio at 10.9% even at the higher swelling temperature of 130 °C (Figure 3b). PAPEK-130 membranes also exhibited low swelling ratios (7.3%) in H_2_SO_4_ (3 M). These results were attributable to synergistic effects of the limited content of hydrophilic groups and highly rigid and hydrophobic backbone, which effectively limit membrane swelling [15,16,40,43]. PAPEK membranes with low swelling would be incredibly beneficial for membrane selectivity and mechanical strength. 

PAPEK membrane microstructures were analyzed using SAXS and, compared with PAPEK-virgin membranes, PAPEK-130 membranes were found to display a broad scattering peak at 0.8–1.9 nm^−1^ (Figure 3c and Appendix A). This indicated that ion transport pathways had been formed [51], which would enable the resulting PAPEK membranes to have low areal resistance. PAPEK membranes showed weak SAXS peaks only at a high swelling temperature (130 °C), probably due to the insignificant phase separation due to the strong rigidity of the adamantane-containing backbone and the low IEC (0.78 mmolg^−1^) of PAPEK membranes. Membranes with low areal resistance would benefit VFB in obtaining high voltage efficiency, which would be highly favorable for high-power-density VFBs. Examination of the areal resistance of these PAPEK membranes showed that the area resistance was clearly affected by the selective swelling temperature because a higher swelling temperature benefited the yield of broad and continuous ion transport pathways [40], which would significantly reduce the areal resistance (enhancing the conductivity) of the membrane (Figure 3d, Appendix A). The areal resistances of PAPEK membranes decreased from 0.57 to 0.22 Ω∙cm^2^ in PAPEK-60 and PAPEK-130 membranes, respectively, which was due to the increased swelling temperature, resulting in wider ion transport pathways [40,42]. These results matched well with ADL, swelling, and SAXS (Figure 3a–c, respectively).

Generally, membrane morphology is closely related to the phase separation kinetics [52], and the morphology of microphase separation affects ionic conductivity [53]. The morphology of PAPEK-virgin and PAPEK-130 membranes showed that, after swelling, the PAPEK-130 membrane exhibited broader surface grains than those of the PAPEK-virgin membrane, which probably resulted from the selective swelling process (Figure 4). Similar phenomena have been observed in other reported studies [40]. These results were consistent with the present SAXS results (Figure 3c).

High cross-mixing of V ions through the membrane in a VFB would lead to an imbalance between positive and negative electrolytes, which would lead to a lower Coulomb efficiency and capacity retention. Therefore, membrane selectivity is vital for identifying appropriate membranes for VFBs [54]. The size of ion transport pathways and charged ion exchange groups significantly influences membrane selectivity. In traditional nano-separation structures, the excessive expansion of ion transport pathways increases swelling and thus greatly increases the permeability of V ions due to high membrane swelling [29]. 

The present PAPEK membranes exhibited increasing V permeability with preparation temperature (Figure 5a), which matched well with the results of ADL and swelling for PAPEK membranes (Figure 3a,b). The permeabilities of PAPEK-60, 90, and 130 membranes were 1.27 × 10^−7^, 2.12 × 10^−7^, and 8.49 × 10^−7^ cm^2^∙min^−1^, respectively. PAPEK membranes exhibited significantly lower permeability than that of Nafion 212 (42.5 × 10^−7^ cm^2^∙min^−1^), which was probably due to synergistic effects from the Donnan exclusion effect [55,56], between positively charged piperazine groups and V ions, and low membrane swelling resulting from the highly rigid and hydrophobic adamantane-containing backbone [15,16,40].

Membrane mechanical properties are vital parameters for VFB. During the charge–discharge cycle, the membrane endures chemical degradation in the electrolyte with strong acid and oxidation and bears mechanical stress due to continuous extrusion caused by battery operation [57]. Generally, improved membrane conductivity can be obtained by increasing the IEC of membranes or ADLs in acid-doped membranes (e.g., PBI membranes), but this comes at the expense of membrane mechanical strength [58]. The present PAPEK membranes exhibited tensile strengths higher than 33.8 MPa, approximately two times higher than Nafion 212 (17 MPa) [15], indicating that PAPEK membranes had sufficient mechanical strength for VFB application (Figure 5b). The tensile strength of the PAPEK-virgin membrane was 47 MPa, while the resulting swelled PAPEK membranes showed tensile strengths varying from 35.3 to 33.8 MPa, probably due to the doped acid and H_2_O in the membrane acting as plasticizers [40]. Moreover, the elongation at the break of swelled PAPEK membranes increased from 13.1 to 16.2%. The Young’s modulus of PAPEK membranes showed a decreasing trend, which was mainly due to the role of acid and water entering the membrane after selective swelling treatment to act as plasticizers. The Young’s modulus decreased gradually with increasing the temperature of selective swelling(Table 2). This result was consistent with the ADL (Figure 3a) and swelling ratio (Figure 3b) of PAPEK membranes.

### 3.1. VFB Performance of PAPEK Membranes

Coulombic efficiency (CE) is the ratio of a battery’s discharge capacity divided by its charge capacity. The cross-mixing of V ions across the membrane would reduce the CE and result in poor VFB performance [59,60]. Here, the VFB performance of PAPEK membranes was investigated at current densities from 80 to 300 mA∙cm^−2^. PAPEK membranes exhibited high CEs ranging from 98 to 99%, significantly higher than Nafion 212 membranes, which varied from 87 to 94% (Figure 6a). These PAPEK membranes showed outstanding CEs due to their low VO^2+^ permeability, which was the result of the synergistic effects of the Donnan exclusion effect (between positively charged piperazine groups and V ions) and low swelling caused by the highly rigid adamantane-containing backbone [15,16,40]. It is worth noting that PAPEK-130 membranes still maintained CEs higher than 98%, thus indicating that these membranes had excellent selectivity.

PAPEK and Nafion 212 membranes exhibited voltage efficiencies (VEs) inversely proportional with increasing current density, mainly due to the rise of Ohmic polarization caused by increased current density (Figure 6b) [61]. PAPEK membranes exhibited high VEs, with PAPEK-130 exhibiting an impressive VE (80.77%) at a high current density of 260 mA∙cm^−2^. To the best of our knowledge, only a few advanced membranes have been reported that show such high VEs, e.g., a thin-film composite membrane [7], indicating that PAPEK-130 membranes were very suitable for high-power-density VFB. In addition, PAPEK-130 exhibited slightly lower VEs than Nafion 212 membranes due to its somewhat higher areal resistance. These results matched well with the present areal resistance results (Figure 3d). 

Energy efficiency (EE) is vital for evaluating energy loss in the charge–discharge process. VFBs with PAPEK-90 and 130 exhibited impressive EEs, at 91.75 and 91.5% at 80 mA∙cm^−2^, respectively (Figure 6c), which were higher than those of many reported high-performance membranes (Figure 6d) [9,10,11,12,13,18,19,22,25,26,27,30,31,52,62,63]. Notably, PAPEK-130 membranes exhibited impressive EEs (83.3% and 80.7% at 200 and 240 mA∙cm^−2^, respectively), which are slightly lower than those of thin-film membranes [7], with 80% being the highest ever reported, at 260 mA cm^−2^, or even better than the few reported outstanding membranes at 200 mA∙cm^−2^ (81.34 [64], 81.93 [22], 80.0 [65], and 80.7% [66]) and porous PBI membranes [67] (EE = 80% at 220 mA∙cm^−2^). The PAEK-60 membrane exhibited significantly lower VEs and EEs than the other PAPEK membranes and Nafion212 membrane due to its high area resistance (0.57 Ω cm^2^). These results suggest that these PAPEK membranes (PAPEK-90 and 130) could work well at a wide range of current densities (80–300 mA∙cm^−2^) and exhibit excellent EEs in high-power-density VFBs.

### 3.2. Stability of PAPEK Membranes

The membrane chemical stability during cycling is one of the most critical challenges in determining whether the membrane is suitable for practical VFB applications [52]. A VFB with a PAPEK-90 membrane exhibited stable battery performance for 2000 cycles at a high current density of 180 mA∙cm^−2^ (Figure 7a). Meanwhile, the discharge capacity of a VFB with a PAPEK-90 membrane was better maintained than that of a VFB with a Nafion 212 membrane at 180 mA∙cm^−2^ current density (Figure 7b and Appendix A).

The stability of PAPEK membranes in 0.15 M VO_2_^+^ in 3 M H_2_SO_4_ solution is summarized in Figure 8. Oxidation of membranes by VO_2_^+^ produces VO^2+^, which can be used as an indicator of membrane oxidation [68]. PAPEK-130 membranes showed a VO^2+^ concentration of 19.9 mmol L^−1^ for 41 days, and the VO^2+^ concentration for Nafion212 membranes was significantly lower than that of PAPEK membranes, mainly due to the high stability of fluorocarbon backbone of Nafion212. PAPEK membranes were significantly lower than SPTES membranes [69] (about 20 mmol L^−1^, 13 days), but also significantly higher than our reported adamantane-containing membranes [39,40,41], which may be due to the poor stability of the hydroxyethylpiperazine groups in oxidative vanadium electrolytes, more related in-depth research is underway in our laboratory.

## 4. Conclusions

New adamantane-based membranes were prepared for VFB application. Basic piperazine and hydrophilic hydroxyl groups were introduced into the highly rigid, hydrophobic, adamantane-containing backbone and selectively swelled in hot phosphoric acid to construct broad and continuous ion transport pathways in the membranes, which exhibited enhanced conductivity, limited swelling, high selectivity, and high mechanical strength. Consequently, VFB with PAPEK membranes exhibited impressive battery performance, e.g., PAPEK-90 membranes yielded CE at 98.7%, VE at 92.9%, EE at 91.7% (80 mA∙cm^−2^), and PAPEK-130 membranes exhibited CE at 99.1%, VE at 81.5%, and EE at 80.7% (240 mA∙cm^−2^). In addition, the present PAPEK membranes exhibited good stability in cycle testing for 2000 cycles. This method of increasing membrane conductivity using selective swelling is versatile and, if suitable membrane materials are used, we believe it can be extended to fabricate high-performance membranes for alkaline fuel cells, organic flow batteries, fuel cells, or water electrolysis.

## Data Availability

The raw data and samples presented in this study are available on request from the corresponding authors. All relevant processed data is shown in the manuscript and Appendix A associated.

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
