# Peer review of "Robust Adamantane-Based Membranes with Enhanced Conductivity for Vanadium Flow Battery Application"

_polymers, 2022, doi:10.3390/polym14081552_

Round 1

Reviewer 1 Report

The authors reported a piperazine substituted PAPEK membrane for vanadium-based redox flow battery applications. It is an interesting work but needs major revision on the characterization of the polymer structure and the test of battery performance.

1) On the structure, the 1H NMR only suggests the disappearance of proton from -CH2Cl group, but doesn't show the proton signals from the substituted piperazine. The authors need to firm up on this, by e.g. running 13C NMR. This is critical because the structure of the polymer has to be confirmed.

2) On the stability of the membranes, the cycling lifetime (Fig. 6b) has shown a significant improvement compared to Nafion212. However, the battery test with Nafion212 must have been somehow compromised because this is orders of magnitude lower than the commercial vanadium flow battery using Nafion membranes. The authors should repeat their experiments to make sure that the battery performance is on par with the standard. 

Reviewer 3 Report

The paper “Robust Adamantane-Based Membranes with Enhanced Conductivity for Vanadium Flow Battery Application” by B. Zhang et al. reports the synthesis and the characterization of Adamantane based membranes to be used in Vanadium flow battery. In particular the obtained membranes show interesting mechanical and conducibility properties as well as high swelling resistance. The work is well organized and presented, the experiments are well planned and the results are clear. Some minor points should be clarified by the authors, and I will list them in the following:

  • It is not clear how the authors measure the area of the membranes and how the error on these measurements affect the evaluation of the swelling ratio.
  • The cycling test is performed only with the PAPEK-90 membrane. This choice should be motivated, indeed it seems that the performances of the PAPEK-130 membrane are very similar or even better.
  • The authors should comment the quite different results obtained with the PAPEK-60 membrane.
  • Future perspective for this research could be also mentioned.
  • The resolution of figure 3 should be improved.

Reviewer 4 Report

The manuscript reported the novel and robust adamantane-based membranes with enhanced conductivity for VRFB applications. Low content basic piperazine and hydrophilic hydroxyl groups are introduced into hydrophobic adamantane containing PAPEK backbone and then selectively swelled to induce microphase separation and form ion transport channels. It is found that the selective swelling temperature has a significant influence on the areal resistance of the resulting membrane. The resulting PAPEK membranes exhibit low swelling, high selectivity, and low areal resistance. In the VRFB single-cell test, the VRFB assembled with these membranes exhibit excellent energy efficiency and stable cycling performance.

I consider the content of this manuscript will definitely meet the reading interests of the readers of the Polymers journal. However, the discussion and explanation should be further improved. Therefore, I suggest giving a minor revision and the authors need to clarify some issues. This could be a comprehensive and meaningful work after revision.

The detailed comments are included in the PDF file.

Round 2

Reviewer 1 Report

I am afraid that the authors didn't answer my second question. The capacity fade rate for the Nafion212 membrane (Fig. 6b) is so much faster than the literature report (e.g. Journal of Membrane Science 580 (2019) 110-116). At such a fast decay rate, the battery will be completely dead after only 200 cycles! VFB using Nafion membrane is a commercialized technology. How could such a fast fade rate be sensible?   

Reviewer 2 Report

The manuscript still needs major modifications, and the authors were not able to respond to most of the questions (which needed further experiments) due to the lack of time. I, therefore, suggest that the editor should give them longer time (more than a month) to let them to respond to the concerns properly.

Here, I am again copying my initial comments (from last round of review) and then my comments based on this round of evaluation of the revised manuscript side-by side:

1. Comment of last round: “The chloromethylated adamantane, containing poly(aryl ether ketone) (CAPEK) with a low degree of chloromethylation (DC) at 0.5 ( 1.0 g), was dissolved in the 1-methyl-2-pyrrolidinone (NMP, 7.3 mL) and N-2-hydroxyethyl piperazine (0.5 g) then added to 95 the solution and stirred at room temperature for 24 h.”- the authors must give the proof against the comment of low degree of chloromethylation (DC).”

Comment after evaluation of this round: Authors have provided the NMR for the degree of chloromethylation (DC), but it still needs proper explanation of the calculation of the DC.

2. Comment of last round: “Chloromethylated adamantane containing poly(aryl ether ketone) (CAPEK) was provided as follows. Adamantane containing poly(aryl ether ketone) (10.0 g) had been dissolved in nitrobenzene (500 mL), then chloromethyl ethyl ether (CMEE) (6 mL) and 2 mL SnCl4 were added to the polymer solution”- the authors should give a reference to prove that the reaction was done in nitrobenzene as chloromethylation can happen in nitrobenzene too.

Comment after evaluation of this round: Answered properly.

3. Comment of last round: “1H NMR analysis of adamantane-based poly(aryl ether ketone) (APEK)and CAPEK were carried out using chloroform-d as the solvent (Fig. 1a), and the degree of substitution (DS) was determined using 1H NMR.” -the authors must provide the process to calculate the degree of substitution (DS) by 1H NMR.

Comment after evaluation of this round: Authors should provide reference for the calculation of the degree of substitution.

4. Comment of last round: Figure 1a (1H NMR of APEK and CAPEK): 1H NMR should be presented with proper integration.

Comment after evaluation of this round: Authors have provided the NMR peaks for the samples, but they need to add proper integration in writing format which shows the number of the protons at each position (similar to standard scientific papers).

5. Comment of last round: “In this study, nanophase-separated structures were induced by the selective swelling process using high concentration H3PO4 (85 wt.%). Because the basic piperazine and hydrophilic hydroxyl grafted in PAPEK membranes had acid-base interactions and formed H-bonds with H3PO4, respectively, these membranes could be selectively swelled by H3”- the authors should provide a proof (TGA thermogram may be) to prove H3PO4 doping.

Comment after evaluation of this round: The authors should have longer time to be able to answer this question, since TGA results are needed before publishing the manuscript.

6. Comment of last round: “1H NMR spectra of the CAPEK polymer were obtained using a Bruker Avance 500 M and CDCl3 as the test solvent.” – the authors should do the basic characterization of the polymers, like molecular weight determination, DSC and TGA.

Comment after evaluation of this round: Authors are still needed to add DSC, TGA, and GPC results to the manuscript. They can add these results and after that they can refer to their previous papers.

7. Comment of last round: In the membrane preparation part, the authors mentioned that they swelled the membranes in H3PO4 and then exchanged with H2SO4. Finally, they stored the membranes in the H2SO4 before using for measurements; however, the time of storing in H2SO4 was not mentioned at the last step. This time can significantly impact the resistivity and many other results if varies from one sample to another.

Comment after evaluation of this round: I agree with the authors that in the longer times, resistivity of the membranes will not be affected significantly by storing time. However, authors are still required to add the time of storing because there should be a difference in the samples which has been stored for 1 day and samples which have been stored for 1 month.

8. Comment of last round: The reviewer does not completely agree with the authors regarding the FTIR data. The authors have mentioned that the peak in FTIR data at ~3339 cm-1 is due to the OH groups of the H2SO4/H2O for PAPEK-90 membrane; however, this peak can be due to the OH groups of the N-2-hydroxyethyl piperazine. Therefore, authors are needed to fix this part of the manuscript before publication.

Comment after evaluation of this round: FTIR results are not showing that OH-groups of sulfuric acid has reacted with PAPEK-90. Therefore, the authors need to justify the reason behind adding the FTIR data to the manuscript.

9. Comment of last round: Based on Figure 2a and Figure 2b, the first swelling in the phosphoric acid at elevated temperature was leading to an increase in the acid doping level and swelling of the samples in the sulfuric acid. The authors need to explain the reason for increasing the ADL and swelling ratio in sulfuric acid which was followed by phosphoric acid and heat treatment.

Comment after evaluation of this round: Answered properly.

10. Comment of last round: The paper uses swelling-induced phase separation where the swelling of the membrane was done at varied temperature (60, 90, 130 C). The reviewer was curious to see the effect of swelling temperature on SAXS data for all temperatures, rather than just at 130 C. Also, the 130 C-swelling led to a broad peak which likely suggested a distribution in ionic domain spacing and size. The SAXS data could have discussed this in addition to reporting the domain spacing values (i.e., d-spacing = 2*pi/q). The domain size/spacing is especially important to know as the authors claim to tune the ionic domains by swelling. The SEM images were not very clear and did not help understand the effect of swelling temperature on phase separation.

Comment after evaluation of this round: The authors should have longer time to be able to answer this question, since SAX results are necessary before publishing the manuscript.

11. Comment of last round: It will be better not to use the term “channel” (like, ion transporting channels), rather use ion transport pathways. Channels are more appropriate for MOF, COF-type systems which creates true ion transporting channels.

Comment after evaluation of this round: Fixed in the text.

12. Comment of last round: The areal resistance of all versions of PAPEK membranes are higher than Nafion suggesting PAPEK membranes are no better than Nafion in terms of proton conductivity. Even though that does not make the paper worthy of publishing some comments/insights on what possibly has caused this (based on prior literature) could enlighten the readers about the potentials or scopes of PAPEK-based materials. Is the low proton conductivity happening because the positive piperazine is also repelling the H+ ion or there is a deep connection with nanostructure? Please convert the membrane areal resistance to conductivity if possible as people are traditionally used to see the conductivity values (in mS/cm).

Comment after evaluation of this round: Answered properly.

13. Comment of last round: The paper mentioned the use of low content of piperazine in the polymer. Although the synthesis section mentioned the mass of piperazine used to make the PAPEK polymers, the number of piperazine units finally attaching to the polymer chains were not quantified (either by NMR or any other technique). This is an important information needed to make the readers understand how low is “low” in this work, i.e., how low a piperazine addition to a polymer chain helped to get the reported performance.

Comment after evaluation of this round: Authors are needed to mention how they are correlating the IEC to the piperazine amount in the samples.

14. Comment of last round: In Figure 1b, there is a typo in the labeling of the graph and based on the text, the orange graph should be assigned to the PAPEK-90.

Comment after evaluation of this round: Still there is discrepancy between the text (Page 4, Line 126) and the Figure 1b. Authors are needed to fix the figure or the text.

Therefore, based on these issues, I believe that the manuscript is still not ready to publish and the authors need longer time to fix the aforementioned concerns.

Round 3

Reviewer 1 Report

I appreciate the literature search done by the authors. My questions have been properly addressed now. I recommend it for publication. Thank you!

Reviewer 2 Report

Figure S3 and S6 containing DSC, TGA and GPC are still missing in the supporting information provided with this round of review request. However, the authors mentioned in the author comments and in the revised manuscript that these data are included in the supporting information. Upon making these revisions, the paper will be acceptable for publication.